# Effects of high-intensity interval training on physical performance, systolic blood pressure, oxidative stress and inflammatory markers in skeletal muscle of spontaneously hypertensive rats

Thaynara Zanoni D'Almeida[1]☯, Mariana Janini Gomes[2]☯, Leticia Estevam Engel[1], Ines Cristina Giometti[1], Natalia Zamberlan Ferreira[2], Rafael Stuani[1,3], Camila Renata Corrêa[4], Robson Chacon Castoldi[5], Sarah Gomes Nunes[1], Andreo Fernando Aguiar[6], Anthony César Castilho[1], Marina Politi Okoshi[4], Francis Lopes Pacagnelli[1]*

1 Postgraduate Program in Animal Science, UNOESTE, Presidente Prudente, Brazil, 2 Department of Kinesiology and Sports Management, Texas A&M University, College Station, TX, United States of America, 3 Postgraduate Program in Health Science, UNOESTE, Presidente Prudente, Brazil, 4 Botucatu Medical School, Sao Paulo State University (UNESP), Botucatu, Brazil, 5 Department of Pharmacology, Institute of Biosciences, Júlio de Mesquita Filho Paulista State University, Botucatu, São Paulo, Brazil, 6 Postgraduate Program in Physical Exercise in Health Promotion, Northern University of Paraná, Londrina, PR, Brazil

☯ These authors contributed equally to this work.
* francispacagnelli@unoeste.br

**Data Availability Statement:** The data are held in a public repository: URLs: https://zenodo.org/

## Abstract

### Aim

To investigate whether high-intensity interval training (HIIT) improves physical performance, systolic blood pressure, and markers of oxidative stress and inflammation in skeletal muscle of spontaneously hypertensive rats (SHR).

### Methods

Nineteen male SHR rats were randomly assigned to two groups: sedentary (SHRC) and trained (SHR+T). The SHR+T group trained five times a week for eight weeks on a treadmill, while the SHR group remained without any exercise stimulus throughout the experimental period. Maximum physical performance and systolic blood pressure (SBP) were assessed before and after the training period. The following variables were measured in the tibialis anterior (TA) muscle: gene expression of the NADPH oxidase complex (NOX2, NOX4, p22$^{phox}$, p47$^{phox}$) and the NF-kB pathway (NF-kB and Ik-B), lipid peroxidation (malonaldehyde; MDA), protein carbonylation, hydrophilic antioxidant capacity (HAC) and pro-inflammatory cytokines (IL-6 and TNF-α).

### Results

SHR+T rats showed higher physical performance and levels of IL-6, and lower SBP and protein carbonylation (p<0.05), compared with SHRC rats. No significant differences (p>0.05) were observed in the other variables.

records/6207938, and DOI 10.5281/zenodo.
6207937.

**Funding:** This study was financed in part by the
Coordenação de Aperfeiçoamento de Pessoal de
Nível Superior - Brasil (CAPES) - Finance Code
001. The funders had no role in study design, data
collection and analysis, decision to publish, or
preparation of the manuscript.

**Competing interests:** The authors have declared
that no competing interests exist.

## Significance

Our results indicate that HIIT is an effective non-pharmacologic strategy to improve physical
performance, reduce SBP, and modulate the skeletal muscle oxidative damage and inflam-
mation in hypertensive rats.

## 1. Introduction

Cardiovascular diseases are the leading cause of death in the world. Arterial hypertension
(AH) affects an estimated 1.28 billion adults worldwide and is among the modifiable risk fac-
tors for cardiovascular diseases [1]. Sedentary lifestyle increases the risk of developing AH [2]
which, if sustained, ultimately, can progress to heart failure (HF), a clinical syndrome charac-
terized by compromised functional capacity, disability, and poor quality of life [3, 4].

The Oxidative stress, defined as an imbalance between oxidant production and antioxidant
defenses that favors the accumulation of oxygen reactive species (ROS), plays an important
role in the pathophysiology of AH and HF. Clinical and experimental studies have demon-
strated an increase in oxidative stress both in the skeletal muscles and at systemic level during
HF [5–8].

The NADPH oxidase (NOX) complex is a major source of ROS in the skeletal muscle,
expressing three isoforms: Nox1, Nox2, and Nox4 [9]. The Nox2 and Nox4 isoforms contrib-
ute to skeletal muscle abnormalities associated with cardiovascular disorders [9]. The NOX2
catalytic subunit forms a complex with p22$^{phox}$ which depends on p47$^{phox}$ and p67$^{phox}$ cytosolic
regulatory subunits linkage being activated. Conversely, even though NOX4 interacts with
transmembrane protein p22$phox$, it differs from other isoforms because it is constitutively
active and independent of regulatory or activator cytosolic proteins [10–12].

At physiological concentrations, ROS play essential roles in physiological cellular processes;
however, sustained high ROS levels induce damage to DNA, proteins, and lipids [13]. Oxida-
tive damage, such as lipid peroxidation and carbonylation of proteins, are important in hyper-
tension resulting in cell damage and dysfunction and, leading to loss of protein function and
damage to muscle contraction, both contributing to worsening of functional capacity [11, 14].
Furthermore, ROS also modulate the nuclear factor kappa B (NF-kB), a transcription factor
that modulates genes expression in diverse cellular processes, including inflammatory pro-
cesses [10, 11, 14]. Changes in inflammatory cytokines (e.g., IL-6 and TNF-α) and oxidative
stress have been shown to be involved in AH-related decline of muscle function [15–17]. Thus,
understanding how these markers are modulated by therapeutic interventions has beneficial
repercussions for patients with AH.

Physical exercise, such as high-intensity interval training (HIIT), is a well-known strategy
to improve functional capacity in hypertensive patients [18, 19]. HIIT consists of alternating
short periods of high intensity exercise (i.e., 85–95% of maximum oxygen consumption,
VO$_2$max) with moderate or low intensity recovery (i.e., 50–60% of VO$_2$ max) [19, 20].

Recent evidence suggests that HIIT improved systolic and diastolic blood pressure in
patients with hypertension and prehypertension, increased flow-mediated vasodilation, and
improved VO$_2$ peak and resting heart rate [18–20]. In others conditions, such as obesity,
aging, and cerebral ischemic, HITT may modulate oxidative stress and inflammation [21–23],
thus improving skeletal muscle function. However, the effects of HIIT on biochemical and
molecular markers of oxidative stress and inflammation in skeletal muscle associated with
hypertension are still poorly understood. It is of interest to exercise physiologists, clinicians,

and the general population to determine how these changes are related to improved physical performance and reduced SBP, since HIIT is a more time-efficient exercise when compared to moderate-intensity continuous training (MICT).

The purpose of this study was to investigate whether HIIT improves physical performance, SBP, and markers of oxidative stress and inflammation in skeletal muscle of spontaneously hypertensive rats (SHR). We hypothesized that HIIT (trained group) would improve physical performance and reduce SBP, concomitantly with the modulation of inflammatory and oxidative stress markers, compared with untrained control group.

## 2. Methods

### 2.1. Ethical approval

This study was approved by the Animal Experiments Ethics Committee (CEUA Protocol 1167–2016) from the University of Western São Paulo, Presidente Prudente, São Paulo, Brazil. The experimental protocols followed the principles of care for laboratory animals formulated by the Brazilian College of Animal Experimentation (COBEA) and are in accordance with the "Guide for the Care and Use of Laboratory Animals" from the Laboratory Animal Research Institute [24].

### 2.2. Animals and experimental groups

We used 19 male Spontaneously Hypertensive Rats (SHR; twelve-month-old), obtained from the Central Animal Hospital of the State University of Campinas (UNICAMP), São Paulo. They were randomly divided into two groups: control SHR (SHRC, n = 9) and trained SHR (SHR+T, n = 10). Animals were housed in standard cages (41x34x16 cm) with three or four rats per cage. Food and water were provided *ad libitum*. Temperature (21–23˚C), relative humidity (50–60%), and light cycle (inverted 12-hour cycle- light from 7 pm to 7 am) were controlled.

### 2.3. Experimental design

The rats in the SHR+T group was submitted to the training protocol for eight weeks as described below. The physical performance test was assessed before and 24 hours after the last HIIT section; a complementary maximal exercise capacity evaluation was performed at the end of the fourth week to adjust the training load. After 48 hours of physical performance test, the SBP was measured. The rats were euthanized after 36 hours of the end period of experimental protocols, and the evaluations of inflammatory and oxidative stress markers were then performed (Fig 1).

### 2.4. Physical performance test

Rats were familiarized for one week with the treadmill (model TK 1—Inbramed, São Paulo, Brazil) at 6 meters/minute with 0% inclination for 10 minutes. After familiarization, maximum exercise capacity was assessed by running on a treadmill at 6 meters/minute with an increase of 3 meters/minute every 3 minutes until exhaustion [25, 26]. Exhaustion was determined when the rats refused to run after manual stimulation or were unable to coordinate steps. Maximal speed and duration of the test were recorded, and total distance covered was calculated. Rats were tested before (week 0), at week 4 for adjustment of training intensity, and at week 8 (24 hours the last HIIT session) [25].

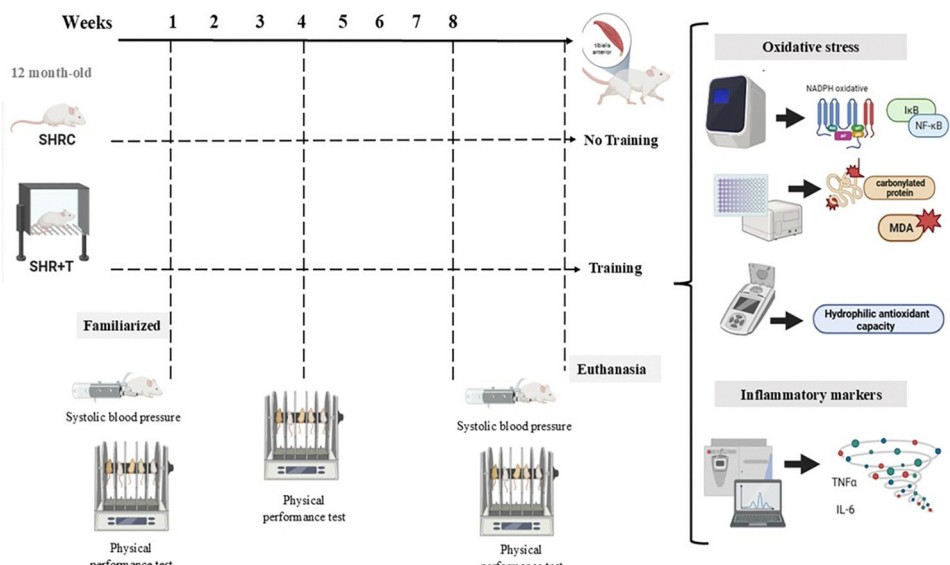

**Fig 1. Schematic figure summarizing the experimental design.**

## 2.5. HIIT protocol

The HIIT protocol was performed for approximately 50 min/day, five days a week, for eight weeks, in an inverted cycle (2 pm to 2:50 pm) [27–29]. Each session consisted of three phases: warm-up, HIIT protocol, and recovery. The warm-up phase included 5 min at 60% of the exhaustion speed, with gradual increases in intensity.

The HIIT protocol phase started at 95% of the speed obtained the exhaustion test for 4 min, alternated with 65% of the maximum speed for 3 min. This HIIT protocol was repeated five times in the first and second weeks (Fig 2). The same velocities of the first week were used in the third and fourth weeks, but the repetitions were increased to six and seven times, consecutively. Before the start of the fifth week, a second physical performance test was performed to reevaluate the maximum velocity of exhaustion; and the training load was adjusted. In the fifth and sixth weeks, HIIT was performed with an adapted protocol at a speed of 23 m/min for 4 min, interspersed at 12 m/min for 3 min, with seven repetitions. Speed was increased 15% in the seventh week and 18% in the eighth week, interspersed with 65% of the maximum speed for 3 min, with seven repetitions.

The recovery phase included 5 min at 60% of the exhaustion speed. The exercise protocol, including details such as intensity, duration, and rest intervals is available in Fig 2.

## 2.6. Systolic blood pressure

Systolic blood pressure (SBP) was determined by plethysmography using the tail-cuff method (Narco Bio-System®, model 709–0610, International medical, Inc., USA.) before and after training protocol. Both assessments pre and post HIIT (48 hours after the physical performance test, 72 hours after the last HIIT session) were performed at the same time of the day, during the dark phase of inverted light cycle (8–9 a.m.), to respect circadian rhythms. Each animal was assessed individually and the average of two SBP readings was recorded [29].

## 2.7. Euthanasia and tissue collection

Animals were euthanized using methods approved for their specific species, stage of development, and size three days after the last HIIT training session. Briefly, the rats were anesthetized

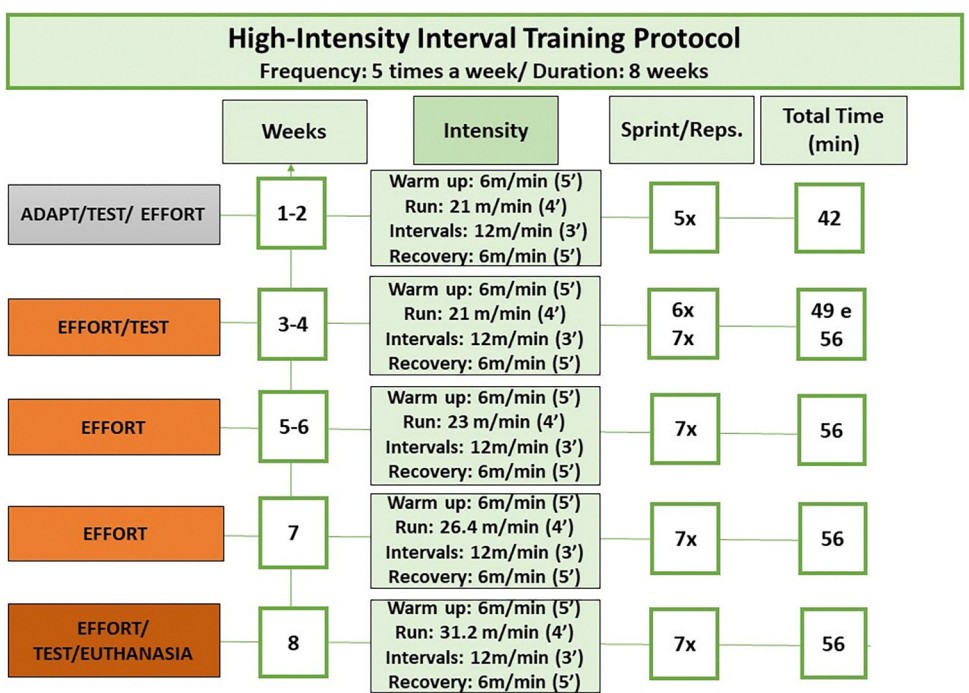

**Fig 2. HIIT protocol.** SBP, systolic blood pressure; HIIT, high intensity interval training; x, times repeated; %, percentage.

with an intramuscular injection of a mixture of ketamine hydrochloride (50 mg/kg) and xylazine hydrochloride (10 mg/kg). Anesthesia levels were assessed by relaxing muscle and mandibular tone and observing the absence of foot reflexes. Euthanasia was performed by decapitation; the procedure was performed by an experienced individual. Samples of tibialis anterior (TA) skeletal muscle were frozen in liquid nitrogen and stored at -80°C [25, 26, 30].

## 2.8. Oxidative stress markers

Tibialis anterior (TA) muscle samples (~100 mg) were homogenized in 1 mL of cold phosphate buffer, pH 7.4. Tissue homogenates were prepared in a motor-driven tissue homogenizer (ULTRA-TURRAX® T25 basicIKA®, Germany). The homogenate was centrifuged at 800 g, for 10 min at 4°C, and the supernatant was assayed for total protein, lipid peroxidation, protein carbonylation, and hydrophilic antioxidant capacity assays. For gene expression, total RNA was extracted from the TA muscle with TRIzol (described below). Analyses of all oxidative stress markers are described below.

**2.8.1 NADPH oxidase complex and NF-kB pathway gene expression.** Relative abundance of mRNA of NADPH oxidase subunits (NOX2, NOX4, p22$^{phox}$, p47$^{phox}$) and the NF-kB pathway (NF-kB and Ik-B) was assessed by real-time polymerase chain reaction after reverse transcription (RT-qPCR). Total RNA was extracted from the TA muscle with TRIzol reagent and treated with DNase I (Invitrogen Life Technologies, Carlsbad, United States), according previously described method [6]. RNA quantification was performed using a spectrophotometer (Nanodrop, Trinean). The RNA purity was considered satisfactory when the ratio between optical densities of 260 and 280 nm was approximately 2.0. One microgram of RNA was reverse transcribed using High-Capacity RNA-to-cDNA Kit (Applied Biosystems, Foster City, United States) with total reaction volume of 20 μl, following the manufacturer's

recommendations. Then, aliquots of 2.5 µL (10–100 ng) of the RT product, containing complementary DNA (cDNA), were submitted to real-time PCR using 10 µL TaqMan $^{TM}$ Fast Advanced Master Mix (Applied Biosystems) and 1 µL of assay (20X) containing sense and antisense primers and Taqman® probes (Applied Biosystems) specific to each gene: *NOX2* (Rn00576710_m1), *NOX4* (Rn00585380_m1), *p22phox* (Rn00577m_m1), *p47phox* (Rn00586945_m1), *NF-kB* (Rn01399572_m1), and *Ik-B* (Rn00584379_m1). Amplification and analysis were performed using the Step One Plus TM Real Time PCR System (Applied Biosystems). Gene expression was normalized by the reference gene cyclophilin (*PPIA;* Rn00690933_m1). Reactions were performed in duplicate and gene expression was calculated using the comparative CT (critical threshold cycle) method (2-ΔΔCT) [31].

**2.8.2 Lipid peroxidation and protein carbonylation.**   Oxidation status of tibial muscle was assessed by lipid peroxidation (malondialdehyde, MDA) and protein carbonylation. Muscle concentration of both oxidant markers was corrected by total protein (measured by the Bradford method).

Malonaldehyde (MDA), a result of the degradation of polyunsaturated lipids, is a marker of lipid peroxidation. MDA (250 µL of muscle tissue supernatant) reacts with TBA, as it is a TBA-reactive substance (TBARS) in the form of 1:2 MDA-TBA. Therefore, the amount of TBARS is proportional to the amount of MDA. The concentration of TBARS was read at 532 nm and calculated using the standard MDA curve and expressed in nmol/mg of protein [32].

The presence of protein carbonyl is a marker of free radical-mediated protein oxidation [33]. Briefly, we combined 100 µL of the tissue supernatant with 100 µL 2,4-dinitrophenylhydrazine (DNPH) (10 mM in 2 M HCl). The samples were incubated for 10 minutes at room temperature, and then 50 µL of NaOH (6 M) was added. It was then incubated again for 10 minutes at room temperature. The reading was performed at 450 nm in a Spectra Max 190 microplate reader (Molecular Devices®), and the results were obtained from the absorbance of the samples and the molar extinction coefficient (22000 M-1 cm-1). The assay result is expressed in nmol/mg of proteins [32].

**2.8.3 Hydrophilic antioxidant capacity (HAC).**   The fluorometric measurement of hydrophilic antioxidant capacity [34] was determined using a microplate reader (VICTOR X2 reader; PerkinElmer, Boston, United States). The antioxidant capacity was quantified by comparing the area under the curve related to the oxidation kinetics of the phosphatidylcholine (PC) suspension, which was used as a reference for the biological matrix. We used 2,2' Azobis (2-amino-propane) -dihdrochlorine (AAPH) as a peroxyl radical initiator. The results represented the percentage of inhibition of (4,4-difluoro-5- (4-phenyl 1–3 butadienyl) -4-bora-3rd, 4th-diaza-s-indacene-3-acidecanoic acid (BODIPY) 581/591 in plasma, with respect to what would occur in the control sample of BODIPY 581/591 in the PC liposome. All analyzes were performed in triplicate. The results are presented by the percentage of protection [34].

## 2.9. Inflammatory markers

Muscle concentrations of the pro-inflammatory cytokines tumor necrosis factor-alpha (TNF-α) and interleukin-6 (IL-6) were quantified in TA samples using ELISA commercial kits (TNF-α: DY510, IL-6: DY506; R&D Systems®, MN, USA). Assay results were detected by the microplate reader Spectra Max 190 (Molecular Devices®, Sunnyvale, United States) [32, 34].

## 2.10. Statistical analysis

The normality of the data was assessed using the Shapiro-Wilk test. Parametric data (physical performance, SBP pro-inflammatory cytokines, gene expression of NOX2, NOX4, p22$^{phox}$, p47$^{phox}$ and NF-kB, hydrophilic antioxidant capacity, lipid peroxidation and protein

carbonylation) were compared by the unpaired t test and is presented as mean ± standard deviation. Non parametric data (gene expression of Ik-B) was compared by the Mann Whitney test and is presented as median, minimum, and maximum. Statistical analysis was performed by JMP software (SAS, Cary Institute, North Carolina, United States) and GraphPad Prism® (GraphPad software, La Jolla, United States). The level of significance was established at 5% (p <0.05).

## 3. Results

### 3.1. Physical performance

By the end of the training program, the SHR+T group covered a greater total distance in the maximum capacity test compared to the SHRC group (1086 ± 140 vs. 280 ± 43 m; $p < 0.0001$).

### 3.2. Systolic blood pressure

Initial SBP was similar between groups, and it was reduced in the SHR+T group after training (Table 1). The rats did not show any signs of heart failure, such as pleural effusion, ascites, tachypnea, and atrial thrombus. No deaths were observed in the SHRC or SHR+T groups.

### 3.3. NADPH oxidase complex and markers of oxidative stress

Gene expression of the NADPH oxidase complex ($p22^{phox}$, $p47^{phox}$, NOX2, and NOX4) and NF-kB pathway (NF-kB and I kB) was similar between SHRC and SHR+T groups ($p>0.05$, Fig 3).

   Protein carbonylation, a marker of protein oxidative damage, was significantly lower in SHR+T than SHRC group (43.94 ± 13,87 nmol/mg vs. 66.04 ± 24,94nmol/mg) $p <0.05$, Fig 4B. MDA concentration, a marker of lipid peroxidation, and hydrophilic antioxidant capacity in TA muscles did not differ between groups (Fig 4A and 4C, respectively)

### 3.4. Assessment of cytokines levels

The SHR+T group had significantly higher muscle concentration of IL-6 compared to the SHRC group (SHRC: 118.0 ± 41,24; SHR+T: 181.3 ± 62,49 pg/ml; $p<0.05$). There were no differences in TNF-α levels between the SHR+T and SHRC groups ($p>0.05$) (Fig 5).

## 4. Discussion

We investigated the effects of HIIT on physical performance, SBP, and markers of oxidative stress and inflammation in skeletal muscle of SHR rats. Our main findings were the following: (1) an 8-week HIIT protocol increased functional capacity in SHR rats and reduced systolic blood pressure (SBP), and (2) HIIT increased IL-6 levels, reduced protein oxidative damage,

**Table 1. Initial and final systolic blood pressure.**

| VARIABLES | SHRC (n = 9) | SHR+T (n = 10) |
|---|---|---|
| Initial SBP (mmHg) | 202 ± 25 | 210 ± 18 |
| Final SBP (mmHg) | 229 ± 5 | 198 ± 18* |

SHRC: control hypertensive (n = 9), SHR+T: trained hypertensive (n = 10), SBP: systolic blood pressure. Data expressed as mean ± standard deviation (Shapiro-Wilk and t nonparametric tests).

* $p <0.05$ vs. SHRC.

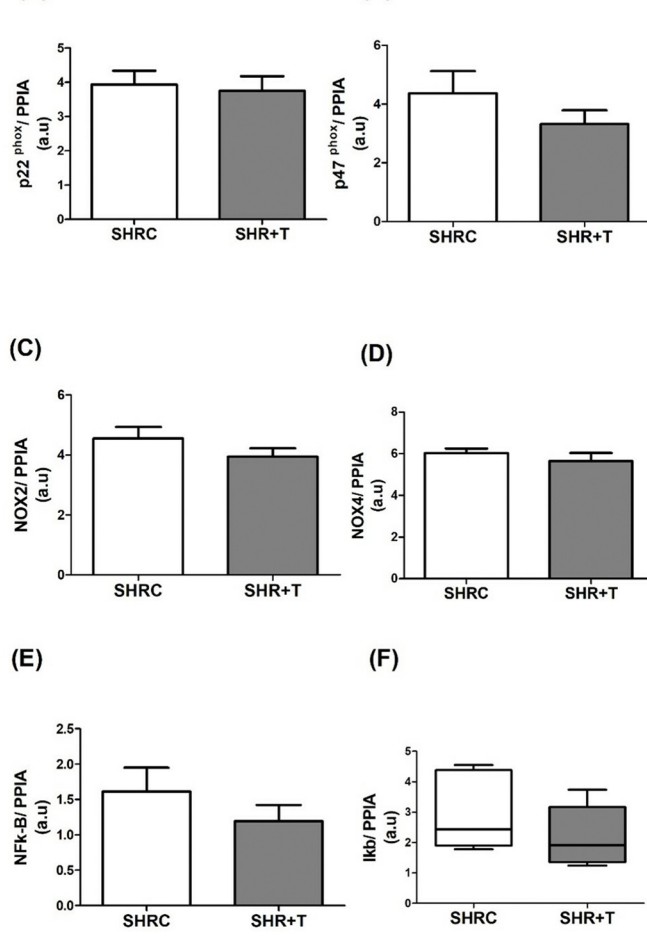

**Fig 3.** Gene expression of NADPH oxidase complex (a-d) and NF-kB pathway (e-f) in TA muscles. SHRC: control hypertensive (n = 9), SHR+T: trained hypertensive (n = 10), a.u.: arbitrary unit. Data expressed as mean ± standard deviation (t test) or median, minimum, and maximum. (Mann Whitney test).

and did not alter gene expression of the NADPH oxidase complex or NF-kB pathway in the TA muscle of SHR rats (Fig 6).

Increasing evidence show that exercise training is beneficial in the treatment of hypertension [35, 36]. The progressive HIIT protocol used in our study promoted a significant improvement in functional capacity, characterized by better exercise tolerance, which is a

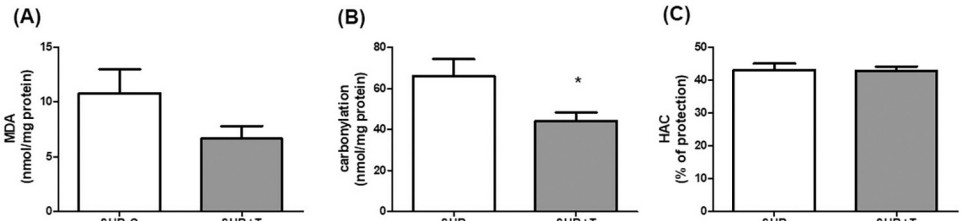

**Fig 4. Oxidative stress in TA muscles.** (a) MDA: malonaldehyde, (b) protein carbonylation, (c) HAC: hydrophilic antioxidant capacity. SHRC: control hypertensive (n = 9), SHR+T: trained hypertensive (n = 10). Data expressed as mean ± standard deviation. t test, *$p < 0.05$ compared to SHRC.

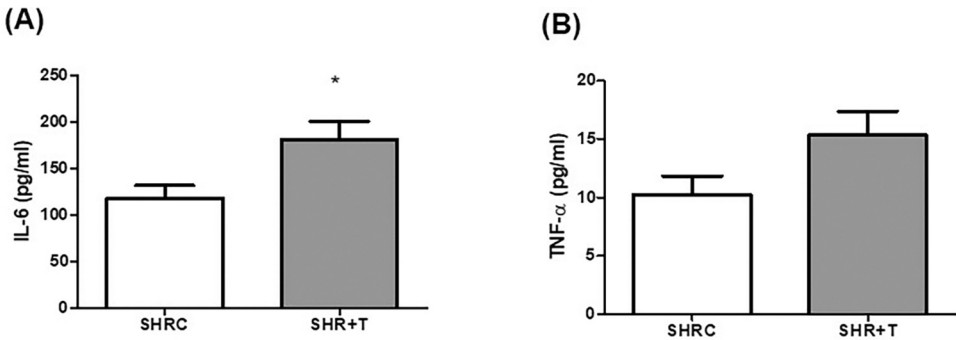

**Fig 5. Cytokine levels in TA muscles.** (a) IL-6: interleukin-6, (b) TNF-α: tumor necrosis factor-alpha. SHRC: control hypertensive (n = 9), SHR+T: trained hypertensive (n = 10). Data expressed as mean ± standard deviation, t test, * p < 0.05 compared to SHRC.

predictor of greater survival [29]. Supervised HIIT effectively improves exercise capacity in patients with stage 1 and 2 hypertension during a short-term exercise program [20]. A systematic review comparing HIIT with MICT found that HIIT led to significantly greater improvements in VO$_2$max, indicating a more substantial enhancement in functional capacity compared to MICT [19]. In our study, 8 weeks of HIIT resulted in a notable increase in functional capacity and a decrease in SBP, underscoring its potential to reduce cardiovascular disease risk factors. A meta-analysis of five studies showed that HIIT significantly reduced blood pressure by an average of -4.7 mmHg (95% CI, -7.7 to -1.8; N = 258) in hypertensive participants compared to a control group; however, the quality of evidence supporting HIIT's effect on systolic and diastolic blood pressure was considered low [20]. Costa et al. compared HIIT and MICT in patients with high-normal blood pressure and hypertension, finding that both exercise methods led to similar reductions in resting blood pressure in these patients [19].

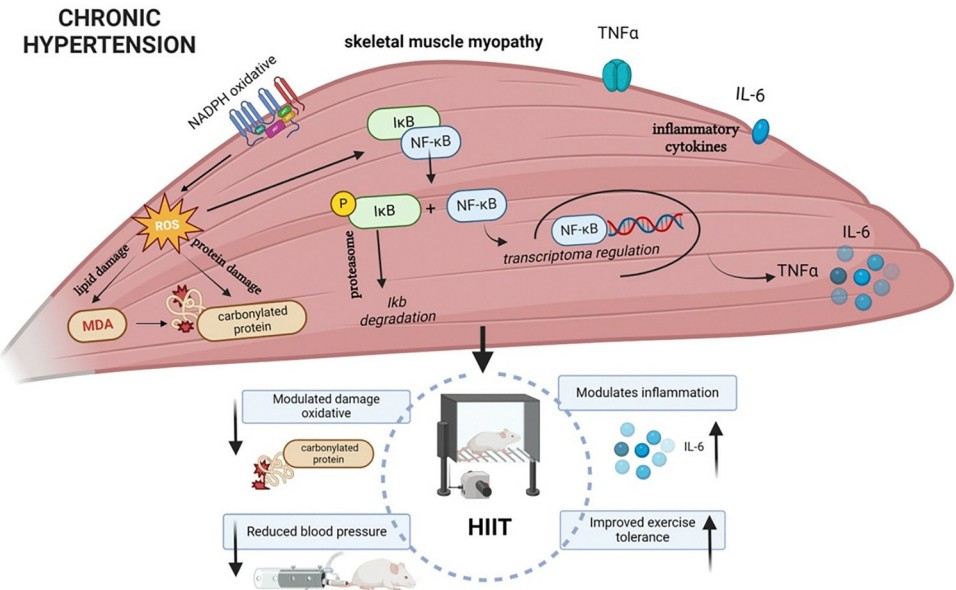

**Fig 6. Schematic representation of the skeletal muscle adaptations to HIIT in rats with chronic hypertension.** Emphasis on the impact of HIIT on muscle oxidative stress and inflammatory markers, highlighting its potential therapeutic implications for cardiovascular health.

To explore the molecular events underlying the HIIT-induced systemic health benefits, we then analyzed the skeletal muscle with a focus on oxidative stress and inflammation. Oxidative stress plays a critical role in the pathogenesis of AH. Experimental models of hypertension have shown an increased oxidative stress and muscle dysfunction in response to sustained hypertension [37].

Muscular contraction stimulates reactive oxygen species (ROS) and inflammatory cytokines production. Exercise-induced increases in the production of ROS and cytokines play an essential role in physiological cellular processes and skeletal muscle adaptation to exercise training. However, prolonged or high intensity exercise can result in adverse effects, such as oxidative damage and impaired muscle regeneration [38]. It is questioned whether ROS could be a determining factor for the inflammatory response or whether exercise-induced adaptive antioxidant effects could scavenger ROS without affecting the inflammatory cascades. Studies have observed that running exercises increased the inflammatory response but did not increase ROS levels [39, 40]. In addition, it has been shown that both prolonged high-intensity and moderate continuous exercises induce oxidative stress and inflammation [41, 42]. However, exercise-induced oxidative stress is determined not only by exercise intensity, but also the type and duration of exercise [43].

Several conditions, including sustained hypertension, can lead to a redox imbalance in skeletal muscle, which favors the accumulation of ROS in the muscle and is associated with loss of muscle mass, increased inflammation, and progression of pathological states such as sarcopenia and muscular dystrophies [16]. The NADPH oxidase family, enzymes collectively referred to as NOX whose function is to generate ROS, plays an important role in muscle physiology [9]. Exercise is a potent modulator of NOX activity. Cunha et. al evaluated plantaris muscle of rats with myocardial infarction-induced HF and demonstrated reductions in NOX2 gene expression after aerobic training. These authors also observed a decreased in ROS production and NF-kB overactivation [44]. In our study, 8 weeks of HIIT protocol did not change NOX2, NOX4, and their subunits gene expression. Intracellular ROS signaling pathways are not completely understood in skeletal muscle of hypertensive rats subjected to exercise.

We first investigated the mRNA levels of NADPH oxidase subunits in the skeletal muscle of SHR rats subjected or not to HIIT. We found no differences between groups, which led us to interpret that HIIT did not affect ROS levels in the skeletal muscle of SHR rats. Additionally, the hydrophilic antioxidant capacity was also measured, which also showed no difference between the groups, demonstrating that the exercise applied in this study did not overproduce ROS, so an increase in antioxidant defense was not expected.

ROS are important actors in exercise-associated adaptations. Malonaldehyde (MDA) and protein carbonylation are markers of oxidative damage [45–47]. Carbonylation of proteins is an irreversible oxidative damage, often leading to a loss of protein function, damage to muscle contraction and worsening of functional capacity [59]. We evaluated the effects of HIIT on oxidative damage by assessing the muscle concentration of protein carbonylation and MDA, markers of protein oxidative damage and lipid peroxidation, respectively. We observed that trained SHR animals had lower concentrations of carbonyl proteins in skeletal muscle than SHRC group, while the levels of MDA in skeletal muscle did not differ between the groups, suggesting that HIIT reduced protein oxidative damage but did not alter oxidative damage to lipids. Since it is known that arterial hypertension promotes oxidative stress, we can conclude that the amount of MDA present in animals from both groups may be reacting directly with muscle protein, favoring carbonylation in this experimental model of AH. These carbonylated proteins are degraded by proteasomes, and it has been noted that proteasome activity can improve with exercise. This may explain the decrease in protein carbonylation in trained animals, associated with improved functional capacity. However, this mechanism requires further studies to be better clarified [48–50].

To the best of our knowledge, there are no studies on the oxidative stress-lowering effects of HIIT in skeletal muscle in the context of hypertension. Although there are evidence that moderate-intensity exercise modulates oxidative stress in skeletal muscle of hypertensive rats, Sánchez and colleagues [51], in a recent study, investigated the effects of moderate intensity exercise on skeletal muscle of hypertensive rats. These authors demonstrated that the soleus and EDL oxidant production increased 139.8% in the muscle of the hypertensive group in comparison to the control group. Regarding the antioxidant systems (total glutathione, GSH content, catalase activity) decreased in the hypertensive. No statistically significant differences were observed between the hypertensive trained group and the hypertensive untrained group in these parameters.

Furthermore, accumulation of ROS can activate the nuclear factor kappa B (NF-kB) pathway, which in turn modulates genes expression in several cellular processes, including inflammatory processes [10, 11, 14]. We then evaluated the mRNA abundance of NF-kB pathway and levels of the cytokines IL-6 and TNF-α in the skeletal muscle. Interestingly, we observed an increase in the concentration of IL-6 in skeletal muscles from the HIIT group, without any changes in TNF-α levels or mRNA levels of the NF-kB pathway. IL-6 is a key myokine produced and released by active skeletal muscles. While the changes in cytokine production by contracting muscles during exercise are well-documented and depend on the exercise's intensity and type [52, 53], there is less information on how regular exercise affects cytokine levels in resting muscles over the long term. Current research shows that regular exercise can alter the cytokine profile, but the extent of these changes is still debated [52].

Regular exercise can lower basal plasma levels of IL-6 and TNF-α, but data on cytokine levels in the resting muscle are limited. On the other hand, increased IL-6 release has been reported during strenuous exercise [54]. Therefore, the observed increase in IL-6 level in the skeletal muscle of our HIIT group can be attributed to the high-intensity nature of the exercise protocol used. Although IL-6 is mainly defined as a proinflammatory cytokine, it has pleiotropic functions in different tissues and organs [54–56]. Evidence has shown that IL-6 may be related to exercise-induced muscle adaptations [53–57], it plays an important role in glucose metabolism in skeletal muscle, and exert its effects in other organs as well, affecting liver and adipose tissue [58, 59]. Muscle-derived IL-6 may also inhibit the effects of pro-inflammatory cytokines such as TNF-α [59]. Thus, increased IL-6 levels could at least partially explain the increased physical performance in our study.

Taken together, our results suggest that HIIT induces skeletal muscle adaptations in SHR rats, likely modulated by a reduction in protein oxidative damage. Understanding the molecular mechanisms underlying the HIIT-induced muscle adaptations is extremely important to inform new or more effective approaches for the treatment of chronic diseases. This study has some limitations that should be mentioned. First, the sample size of our study is small, which may increase the chance of type I error. Second, the animal model of spontaneously hypertensive rats does not necessarily translate the effects of hypertension associated with lifestyle in humans, which prevents us from establishing robust conclusions about clinical application. Finally, other inflammatory markers (e.g., IL-10 and IL-15) and oxidative stress were not analyzed to corroborate our findings. Further studies are needed to address these limitations and confirm our findings.

## 5. Conclusion

Our findings show that HITT is an effective non-pharmacologic strategy for the treatment of chronic hypertension, by reducing systolic blood pressure and improving physical performance, likely via modulation of markers of inflammation and oxidative stress in skeletal muscle.

## Supporting information

**S1 Dataset.**
(XLSX)

## Acknowledgments

The authors would like to thank Eric Schoeffel for translating the article into English.

## Author Contributions

**Conceptualization:** Thaynara Zanoni D'Almeida, Mariana Janini Gomes, Leticia Estevam Engel, Natalia Zamberlan Ferreira, Robson Chacon Castoldi, Anthony César Castilho, Marina Politi Okoshi, Francis Lopes Pacagnelli.

**Data curation:** Natalia Zamberlan Ferreira, Camila Renata Corrêa, Sarah Gomes Nunes.

**Formal analysis:** Thaynara Zanoni D'Almeida, Leticia Estevam Engel, Natalia Zamberlan Ferreira, Anthony César Castilho.

**Funding acquisition:** Marina Politi Okoshi.

**Investigation:** Rafael Stuani, Robson Chacon Castoldi, Sarah Gomes Nunes.

**Methodology:** Thaynara Zanoni D'Almeida, Ines Cristina Giometti, Rafael Stuani, Camila Renata Corrêa, Robson Chacon Castoldi, Sarah Gomes Nunes, Andreo Fernando Aguiar.

**Project administration:** Camila Renata Corrêa, Marina Politi Okoshi, Francis Lopes Pacagnelli.

**Resources:** Ines Cristina Giometti.

**Supervision:** Ines Cristina Giometti, Rafael Stuani, Camila Renata Corrêa, Andreo Fernando Aguiar, Anthony César Castilho.

**Validation:** Andreo Fernando Aguiar.

**Writing – original draft:** Mariana Janini Gomes, Leticia Estevam Engel.

**Writing – review & editing:** Mariana Janini Gomes, Leticia Estevam Engel, Francis Lopes Pacagnelli.

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
