## [Decision Letter · Decision Letter 0]

23 Jun 2024

PONE-D-24-05118High-Intensity Interval Training Modulates Oxidative Damage and Inflammation in Skeletal Muscle of Spontaneously Hypertensive RatsPLOS ONE

Dear Dr. Pacagnelli,

Thank you for submitting your manuscript to PLOS ONE. After careful consideration, we feel that it has merit but does not fully meet PLOS ONE’s publication criteria as it currently stands. Therefore, we invite you to submit a revised version of the manuscript that addresses the points raised during the review process.

We look forward to receiving your revised manuscript.

Kind regards,

Zhiwen Luo

Academic Editor

PLOS ONE

https://www.researchgate.net/publication/348621495_Effect_of_Running_Exercise_on_Oxidative_Stress_Biomarkers_A_Systematic_Review

https://www.hindawi.com/journals/omcl/2023/9979397/

In your revision ensure you cite all your sources (including your own works), and quote or rephrase any duplicated text outside the methods section. Further consideration is dependent on these concerns being addressed.

“This study was financed in part by the Coordenação de Aperfeiçoamento de Pessoal de Nível Superior - Brasil (CAPES) - Finance Code 001.”

Additional Editor Comments:

Thank you for submitting your manuscript to the Journal and as voucan see that the reviewer thinkyour manuscript is interesting and provide valuable comments for youlreference. Please submit the revised manuscript ASAP and also include a rebuttal that would clearly list all the responses to the reviewer's comments.

Reviewers' comments:

Reviewer's Responses to Questions

**Comments to the Author**

1. Is the manuscript technically sound, and do the data support the conclusions?

Reviewer #1: Partly

Reviewer #2: Partly

Reviewer #3: Yes

2. Has the statistical analysis been performed appropriately and rigorously? 

Reviewer #1: Yes

Reviewer #2: Yes

Reviewer #3: Yes

3. Have the authors made all data underlying the findings in their manuscript fully available?

Reviewer #1: Yes

Reviewer #2: Yes

Reviewer #3: No

4. Is the manuscript presented in an intelligible fashion and written in standard English?

Reviewer #1: Yes

Reviewer #2: No

Reviewer #3: Yes

5. Review Comments to the Author

Reviewer #1: The study investigates the impact of High-Intensity Interval Training (HIIT) on oxidative damage and inflammation in the skeletal muscle of spontaneously hypertensive rats (SHR). It aims to understand if HIIT can enhance physical performance by modulating NF-kB signaling, NADPH oxidase expression, and markers of oxidative stress and inflammation in the SHR skeletal muscle. Nineteen SHR rats were randomly assigned to sedentary (SHRC) and trained (SHR+T) groups. The SHR+T group underwent HIIT five times a week for eight weeks on a treadmill, while the SHRC group remained sedentary. Various parameters including systolic blood pressure (SBP), maximum exercise performance, pro-inflammatory cytokines, oxidative stress markers, and antioxidant capacity were measured in the tibialis anterior (TA) muscle. The trained SHR rats exhibited reduced SBP, improved muscle performance, elevated levels of IL-6, and decreased protein carbonylation. However, no significant effects were observed in other oxidative stress markers.

Review Comments:

1. The structure of the manuscript needs improvement for better clarity and flow. Consider revising the organization of sections to enhance readability.

2. The title needs to be more specific and informative. It should clearly indicate the focus of the study and the experimental model used.

3. Provide a more comprehensive background on HIIT and its potential benefits in the context of hypertension. Additionally, clarify the rationale behind choosing specific markers for oxidative stress and inflammation.

4. Expand the description of the HIIT protocol, including details such as intensity, duration, and rest intervals, to facilitate reproducibility. Clarify the rationale behind the selection of the markers measured in the TA muscle and justify why other potential markers were not included.

5. Ensure that the results are presented clearly and concisely. Consider using tables or figures to organize and present the data effectively. Provide statistical details for all comparisons made between the SHRC and SHR+T groups.

6. Discuss the implications of the findings in the context of existing literature on exercise interventions for hypertension. Address potential limitations of the study, such as the small sample size and the focus on a specific rat model.

Reviewer #2: In the study, the authors investigated whether high intensity interval training (HIIT) improves physical performance by modulating NF-kB 33 signaling, the expression of NADPH oxidases, and markers of oxidative stress and inflammatory mediators in the skeletal muscle of the male spontaneously hypertensive rats (SHR). Based on the findings, the authors conclude that HIIT provided systemic benefits, such as reduced blood pressure and improved physical performance, in combination with modulation of skeletal muscle oxidative damage and inflammation.

Major Comments:

1) At what time of day was the SBP measurement conducted? Did all rats undergo measurements at the same time of day? Were both assessments (pre and post) made at the same time? This is important information that should be detailed by the authors.

2) “Each animal was individually coupled to system and the average of two readings was recorded for each meansurement. The SBP was measured from six consecutive cycles per day”. I don't understand well this part. Could the authors explain better this topic?

3) The SBP measurement at the end of the experiment on the following day (hours later) after the maximal exercise test seems to be a problem. The SBP value at the end of the experiment should be assessed on the day following the completion of the 8 weeks of exercise training, with the exercise test conducted afterward. Therefore, it appears that there is a direct influence of hypotension induced by the maximal exercise test on the SBP values.

4) Are the data normally distributed? Why was the Mann Whitney test used?

5) “In the present study, we investigated the effects of high intensity interval training (HIIT) in skeletal muscle abnormalities induced by arterial hypertension in rats”. Do the authors consider the values of the observed parameters abnormal? Perhaps the word "abnormality" may not be the best way to describe the evaluated parameters.

6) A point not adequately discussed by the authors is how the parameters could explain the increase in exercise capacity. Also, it remains unclear what the information about IL-6 should explain.

Minor Comments:

1) Line 133: What stimulation was used?

2) Figure 2 was wrote before figure 1. It is wrong. Please, correct it.

3) The authors reported no signs of heart failure in the rats. Was any histological or echocardiographic evaluation performed? Although the mentioned signs were not verified in the article, there is a possibility that the rats already presented cardiac structural changes, fibrosis, and collagen deposition.

4) There are several sentences containing punctuation error. Also, there are excessive punctuation marks (for example, two periods at the end of a sentence), absence of periods, misspellings, among others. I suggest to perform a spelling review on the manuscript.

5) Several abbreviated terms in the article appear again written out in full later on. I suggest that the authors review this. Additionally, ensure that the use of abbreviations is preceded by the full term earlier in the text. For instance, in the abstract (and article in general), there are abbreviations without the corresponding full terms.

Reviewer #3: The manuscript is well presented and aims to elucidate the effects of an unconventional non-pharmacological treatment for hypertension on systolic blood pressure, physical capacity, inflammation, and oxidative stress. The research provides solid data supporting the proposed conclusions, along with appropriate statistical analysis. However, the individual data sets from the authors were not located. Overall, the manuscript is cohesive and concise, written in proper English.

Suggestions:

a) The sequence of presenting the methods and results of the work should always be consistent across all sections of the article. This facilitates reading;

b) In lines 101 to 105, regarding the hypothesis of the study, there is no mention of blood pressure. Isn't this an important consideration for the experimental model?;

c) In line 139, remove "is shown in Figure 2" is wrong and repeated;

d) In the item "2.4 HIIT protocol," please mention the details of the remaining training weeks;

e) In lines 336 and 337, the manuscript states that "IL-6 plays a role in the adaptations related to training and performance and the anti-inflammatory benefits of exercise." Please explain the hypothesis of how IL-6 contributes to the anti-inflammatory benefits of exercise;

f) In line 368, clarify that the MDA did not change;

g) In lines 373 and 374, "hydrophilic antioxidant capacity, which was not altered by HIIT exercise" please explain why;

h) In lines 374 and 376, "Altogether, our results indicate that HIIT had a significant effect in decreasing excessive ROS that potentially overwhelms antioxidant defenses, leading to oxidative stress in hypertensive conditions" make it clear that this refers to an increase in carbonyls;

i) In line 393, "a reduction of muscle oxidative damage" Please be more specific regarding the findings;

j) Please reconsider the title (blood pressure can be highlighted) and conclusion, focusing only on the findings of the present study without generalizations.

6. PLOS authors have the option to publish the peer review history of their article (what does this mean?). If published, this will include your full peer review and any attached files.

Reviewer #1: No

Reviewer #2: No

Reviewer #3: No

---

## [Author Response · Author response to Decision Letter 0]

29 Nov 2024

Thank you for the opportunity to submit a revised version of our manuscript. We appreciate the time and effort that you have dedicated to providing your valuable feedback on our work. We are grateful to the reviewers for their insightful comments. We have been able to incorporate changes to reflect most of the suggestions provided by the reviewers. We have highlighted the changes in the revised manuscript.

Here is a point-by-point response to the reviewers’ comments and concerns.

We look forward to hearing from you in due course.

Yours sincerely,

Dra Francis Lopes Pacagnelli

Reviewer #1:

Thank you for the opportunity to revise our manuscript. We are grateful for your careful and considered comments. We have made every attempt to respond to each (see below) and to fully address these comments in the revised manuscript and we believe these revisions have resulted in a significantly improved manuscript. 

1. The structure of the manuscript needs improvement for better clarity and flow. Consider revising the organization of sections to enhance readability.

RESPONSE: We would like to thank the reviewer for their careful review of our manuscript and valuable comments which substantially helped to improve the quality of our paper. We apologize for the lack of organization of sections of the previous version. We have added the following sections to the revised manuscript to improve clarity: 2.1. Ethical approval, 2.3. Experimental design. We reviewed the organization of sections. 

2. The title needs to be more specific and informative. It should clearly indicate the focus of the study and the experimental model used.

RESPONSE: Our previous title indicated the focus of the study (the effects of HIIT on the skeletal muscle, specifically oxidative stress and inflammation) and the experimental model used (spontaneously hypertensive rats). However, we have modified the title to “Effects of high-intensity interval training on physical performance, systolic blood pressure, oxidative stress and inflammatory markers in skeletal muscle of spontaneously hypertensive rats” to enhance clarity.

*Previous title: High-Intensity Interval Training modulates oxidative damage and inflammation in skeletal muscle of spontaneously hypertensive rats

3. Provide a more comprehensive background on HIIT and its potential benefits in the context of hypertension. Additionally, clarify the rationale behind choosing specific markers for oxidative stress and inflammation.

RESPONSE: We have addressed to this comment on page 3-4 of the revised manuscript. 

4. Expand the description of the HIIT protocol, including details such as intensity, duration, and rest intervals, to facilitate reproducibility. 

RESPONSE: We have detailed the HIIT protocol in the revised manuscript and have also updated the figure with more details (page 5-6; Figure 2). 

5. Clarify the rationale behind the selection of the markers measured in the TA muscle and justify why other potential markers were not included.

RESPONSE: Oxidative stress is characterized by an imbalance between the production of reactive oxygen species (ROS) and the body’s antioxidant defenses, which can neutralize or repair the oxidative damage that sustained high levels of ROS can cause to lipids, proteins, and DNA. In summary, the markers measured in our manuscript provide a comprehensive view of oxidative stress by indicating the production of ROS (NADPH oxidase), the damage caused by ROS to lipids (MDA) and to proteins (protein carbonylation), and the body’s ability to neutralize ROS (hydrophilic antioxidant capacity). An increase in the oxidant production and oxidative damage and/or a decrease in the antioxidant capacity favors oxidative stress.

Although there are several sources of ROS, such as NADPH oxidase, mitochondria, xanthine oxidase, etc., in the contracting skeletal muscle, NADPH oxidase is the main source of ROS. Skeletal muscles express three isoforms of NADPH oxidases (Nox1, Nox2, and Nox4); the isoforms NOX2 and NOX 4 are the most studied and characterized in the context of exercise and oxidative stress.

In addition, the presence of ROS can be inferred by their effects on protein, carbohydrates, nucleic acids, and lipids to generate specific compounds which, so long as they cannot be formed by other mechanisms, can be used as biomarkers of oxidative damage. We used one of the most assessed markers of lipid peroxidation, malondialdehyde (MDA), and a common protein oxidative marker, which is the formation of protein carbonyls due to oxidation of specific amino acid residues. 

Antioxidants include enzymes and small molecules that react with individual ROS to decrease oxidative damage. The enzymatic antioxidant system primarily includes enzymes like superoxide dismutase (SOD), catalase, and glutathione peroxidase (GPX). These enzymes work to neutralize reactive oxygen species (ROS) and protect cells from oxidative damage.

Hydrophilic antioxidants, on the other hand, are non-enzymatic antioxidants that are soluble in water and can scavenge free radicals and ROS directly. They include vitamins such as vitamin C, as well as other molecules like glutathione and certain flavonoids. These antioxidants complement the enzymatic system by providing additional protection against oxidative stress, especially in aqueous environments within the cell. We evaluated only the hydrophilic antioxidant capacity because we had insufficient tissue sample to measure the activity of the antioxidant enzymes SOD, GPX, and catalase.

6. Ensure that the results are presented clearly and concisely. Consider using tables or figures to organize and present the data effectively. Provide statistical details for all comparisons made between the SHRC and SHR+T groups.

RESPONSE: We have reviewed all data and additional statistical details were added to the revised manuscript (Table 1 legend, page 10; figures 3, 4 and 5 legends, pages 10-11).

7. Discuss the implications of the findings in the context of existing literature on exercise interventions for hypertension. Address potential limitations of the study, such as the small sample size and the focus on a specific rat model.

 RESPONSE: We have included studies of existing literature on exercise and hypertension to the discussion of our findings (page 11). In addition, potential limitations of the study were included (page 14).

Reviewer #2: 

1) At what time of day was the SBP measurement conducted? Did all rats undergo measurements at the same time of day? Were both assessments (pre and post) made at the same time? This is important information that should be detailed by the authors.

RESPONSE: The SBP measurement was conducted during the dark phase of the dark/light cycle (between 8 and 9 a.m.), to respect circadian rhythms, and all rats underwent the measurements at the same time of day. Both assessments pre and post HIIT (72 hours after the last HIIT session) were performed at the same time. This information was included in the revised manuscript (page 6)

2) “Each animal was individually coupled to system and the average of two readings was recorded for each meansurement. The SBP was measured from six consecutive cycles per day”. I don't understand well this part. Could the authors explain better this topic?

RESPONSE: We apologize for this unfortunate mistake on page 6 and the confusion it may have caused the reviewer. We have not measured SBP for six consecutive cycles per day, what was performed and considered to the results was the average of two readings recorded at the same time and day. We have reformulated this description on page 6 of the revised manuscript to enhance clarity and reproducibility. 

3) The SBP measurement at the end of the experiment on the following day (hours later) after the maximal exercise test seems to be a problem. The SBP value at the end of the experiment should be assessed on the day following the completion of the 8 weeks of exercise training, with the exercise test conducted afterward. Therefore, it appears that there is a direct influence of hypotension induced by the maximal exercise test on the SBP values.

RESPONSE: We apologize for missing such important information. The exercise test was carried out 24 hours after the last HIIT session and the SBP measurement was carried out 48 hours after the test (physical performance test), and 72 hours after the last HIIT session, to ensure that we were not evaluating acute responses to endurance exercise. These important details have been added to the text (see page 6).

4) Are the data normally distributed? Why was the Mann Whitney test used?

RESPONSE: The data is normally distributed except by the gene expression of Ik-B, which presented a non-parametric distribution and was analyzed using the Mann Whitney test. We have reformulated the statistics section in the revised manuscript to improve clarity (page 9).

5) “In the present study, we investigated the effects of high intensity interval training (HIIT) in skeletal muscle abnormalities induced by arterial hypertension in rats”. Do the authors consider the values of the observed parameters abnormal? Perhaps the word "abnormality" may not be the best way to describe the evaluated parameters.

RESPONSE: We agree with the reviewer. We have removed the word “abnormality” and rewritten this sentence in the revised manuscript (see page 11).

6) A point not adequately discussed by the authors is how the parameters could explain the increase in exercise capacity. Also, it remains unclear what the information about IL-6 should explain.

Response: As suggested by the reviewer, we have added and discussed further these points in the revised manuscript (pages 12,13 and 14). We appreciate the reviewer’s suggestion.

Minor Comments:

7) Line 133: What stimulation was used? 

Response: Manual stimulation was used; this information was added.

8) Figure 2 was wrote before figure 1. It is wrong. Please, correct it.

Response: We apologize for this error. It has been corrected in the revised manuscript.

9) The authors reported no signs of heart failure in the rats. Was any histological or echocardiographic evaluation performed? Although the mentioned signs were not verified in the article, there is a possibility that the rats already presented cardiac structural changes, fibrosis, and collagen deposition.

Response: In this study we did not perform histological and echocardiographic evaluation. However, other studies (below) with the same model have demonstrated that SHR rats develop heart failure later, from 18 months of age. 

DOI: 10.1016/j.ijcard.2016.04.101

DOI: 10.1016/j.ijcard.2012.03.063

10) There are several sentences containing punctuation error. Also, there are excessive punctuation marks (for example, two periods at the end of a sentence), absence of periods, misspellings, among others. I suggest to perform a spelling review on the manuscript. 

Response: The text was revised, and errors corrected. We appreciate the reviewer’s careful review of our manuscript.

11) Several abbreviated terms in the article appear again written out in full later on. I suggest that the authors review this. Additionally, ensure that the use of abbreviations is preceded by the full term earlier in the text. For instance, in the abstract (and article in general), there are abbreviations without the corresponding full terms.

Response: The text was revised, and errors corrected. We appreciate the reviewer’s careful review of our manuscript.

Reviewer #3: 

The manuscript is well presented and aims to elucidate the effects of an unconventional non-pharmacological treatment for hypertension on systolic blood pressure, physical capacity, inflammation, and oxidative stress. The research provides solid data supporting the proposed conclusions, along with appropriate statistical analysis. However, the individual data sets from the authors were not located. Overall, the manuscript is cohesive and concise, written in proper English.

The datasets were attached at the time of submission, on the journal's website 

Suggestions:

1) The sequence of presenting the methods and results of the work should always be consistent across all sections of the article. This facilitates reading;

Response: We would like to thank the reviewer for their careful review of our manuscript and valuable comments which substantially helped to improve the quality of our paper. We apologize for the lack of organization of sections of the previous version. We have added the following sections to the revised manuscript to improve clarity: 2.1. Ethical approval, 2.3. Experimental design. We reviewed the organization of sections. 

2) In lines 101 to 105, regarding the hypothesis of the study, there is no mention of blood pressure. Isn't this an important consideration for the experimental model?

Response: We agree with the reviewer. The hypothesis was restructured

3) In line 139, remove "is shown in Figure 2" is wrong and repeated;

Response: We apologize for the error. We have corrected this. 

4) In the item "2.4 HIIT protocol," please mention the details of the remaining training weeks;

Response: We have detailed the HIIT protocol in the revised manuscript and have also updated the figure with more details (page 5-6; Figure 2).

5) In lines 336 and 337, the manuscript states that "IL-6 plays a role in the adaptations related to training and performance and the anti-inflammatory benefits of exercise." Please explain the hypothesis of how IL-6 contributes to the anti-inflammatory benefits of exercise;

Response: The hypothesis of how IL-6 contributes to the anti-inflammatory benefits of exercise was added to the revised manuscript. In summary, IL‐6 appears to be a factor in exercise‐mediated activation of AMPK, a cellular energy sensor, which increases fat and glucose metabolism within skeletal muscle fibers, liver, and adipose tissue. Additionally, AMPK activation promotes long-term benefits like mitochondrial biogenesis and improved aerobic capacity. IL-6 released by muscle cells also interacts with immune cells such as monocytes and lymphocytes, triggering anti-inflammatory responses, such as IL-10 release. Overall, IL-6 signaling appears to contribute to both exercise performance improvements and the anti-inflammatory effects of exercise. 

References:

Petersen AM, Pedersen BK. The role of IL-6 in mediating the anti-inflammatory effects of exercise. J Physiol Pharmacol. 2006 Nov;57 Suppl 10:43-51.

Nash et al. IL-6 signaling in acute exercise and chronic training: Potential consequences for health and athletic performance. Scand J Med Sci Sports. 2023 Jan;33(1):4-19. 

6) In line 368, clarify that the MDA did not change;

Response: We apologize for missing such important information. The paragraph has been rewritten details (page 13).

7) In lines 373 and 374, "hydrophilic antioxidant capacity, which was not altered by HIIT exercise" please explain why;

Response: We added this information. The paragraph has been rewritten.

8) In lines 374 and 376, "Altogether, our results indicate that HIIT had a significant effect in decreasing excessive ROS that potentially overwhelms antioxidant defenses, leading to oxidative stress in hypertensive conditions" make it clear that this refers to an increase in carbonyls;

Response: We apologize for missing such important information. We added this information. The paragraph has been rewritten (Page 13).

9) In line 393, "a reduction of muscle oxidative damage" Please be more specific regarding the findings;

Response: The paragraph has been rewritten.

10) Please reconsider the title (blood pressure can be highlighted) and conclusion, focusing only on the findings of the present study without generalizations.

Response: Our previous title indicated the focus of the study (the effects of HIIT on the skeletal muscle, specifically oxidative stress and inflammation) and the experimental model used (spontaneously hypertensive rats). However, we have modified the title to “Effects of high-intensity interval training on physical performance, systolic blood pressure, oxidative stress and inflammatory markers in skeletal muscle of spontaneously hypertensive rats” to enhance clarity.

*Previous title: High-Intensit

---

## [Editor Report · Decision Letter 1]

11 Dec 2024

Effects of high-intensity interval training on physical performance, systolic blood pressure, oxidative stress and inflammatory markers in skeletal muscle of spontaneously hypertensive rats

PONE-D-24-05118R1

Dear Dr. Pacagnelli,

We’re pleased to inform you that your manuscript has been judged scientifically suitable for publication and will be formally accepted for publication once it meets all outstanding technical requirements.

Kind regards,

Zhiwen Luo

Academic Editor

PLOS ONE
---

## [Editor Report · Acceptance letter]

19 Dec 2024

PONE-D-24-05118R1 

PLOS ONE

Dear Dr. Pacagnelli, 

I'm pleased to inform you that your manuscript has been deemed suitable for publication in PLOS ONE. Congratulations! Your manuscript is now being handed over to our production team.

Kind regards, 

on behalf of

Dr. Zhiwen Luo 

Academic Editor

PLOS ONE